# The structure of the COPII transport-vesicle coat assembled on membranes

Giulia Zanetti[1†], Simone Prinz[2], Sebastian Daum[3], Annette Meister[3], Randy Schekman[1,4], Kirsten Bacia[3], John AG Briggs[2]*

[1]Department of Molecular and Cell Biology, University of California, Berkeley, Berkeley, United States; [2]Structural and Computational Biology Unit, European Molecular Biology Laboratory, Heidelberg, Germany; [3]HALOmem, Martin Luther University of Halle-Wittenberg, Halle, Germany; [4]Howard Hughes Medical Institute, University of California, Berkeley, Berkeley, United States

**Abstract** Coat protein complex II (COPII) mediates formation of the membrane vesicles that export newly synthesised proteins from the endoplasmic reticulum. The inner COPII proteins bind to cargo and membrane, linking them to the outer COPII components that form a cage around the vesicle. Regulated flexibility in coat architecture is essential for transport of a variety of differently sized cargoes, but structural data on the assembled coat has not been available. We have used cryo-electron tomography and subtomogram averaging to determine the structure of the complete, membrane-assembled COPII coat. We describe a novel arrangement of the outer coat and find that the inner coat can assemble into regular lattices. The data reveal how coat subunits interact with one another and with the membrane, suggesting how coordinated assembly of inner and outer coats can mediate and regulate packaging of vesicles ranging from small spheres to large tubular carriers.

*For correspondence: john.briggs@embl.de

†Present address: Department of Biological Sciences, Institute of Structural and Molecular Biology, London, United Kingdom

## Introduction

In eukaryotic cells, newly synthesized proteins are transported from the endoplasmic reticulum (ER) to the Golgi apparatus through the action of the coat protein complex II (COPII). Assembly of COPII coat proteins on the membrane leads to generation of coated membrane vesicles carrying cargo molecules. Vesicle formation proceeds via sequential assembly of the coat components. It is initiated by the small GTPase, Sar1. Upon exchange of GDP for GTP (catalysed by Sec12), Sar1 exposes an N-terminal amphipathic helix that inserts into the outer ER membrane leaflet, promoting curvature (*Lee et al., 2005*). Sar1 recruits heterodimers of Sec23 and Sec24 to the membrane, thereby forming the inner layer of the COPII coat (*Matsuoka et al., 1998*). Sec23/24 is an adaptor complex: Sec24 binds transport cargo while Sec23 interacts with Sar1 and recruits Sec13/31 (*Miller et al., 2002*; *Bi et al., 2007*). Sec13/31 heterotetramers constitute the outer coat layer, thought to polymerise into cages that enclose the budding membrane (*Fath et al., 2007*; *Stagg et al., 2008*). GTP hydrolysis on Sar1, activated by Sec23 and further accelerated by Sec31, completes the cycle by promoting fission of the bud and coat depolymerization (*Zanetti et al., 2012*).

X-ray crystallography has been used to obtain structural models for all the coat subunits (*Bi et al., 2002*; *Fath et al., 2007*). Available structural data for the inner coat is limited to isolated subcomplexes. Progress has been made in understanding how the outer coat subunits assemble into a coat by using single-particle cryo-electron microscopy to derive models of Sec13/31 cages formed in vitro under high salt conditions in the absence of membrane (*Stagg et al., 2006, 2008*; *Bhattacharya et al., 2012*; *Noble et al., 2012*). A comparison of the vertices and edges in cages of different sizes (60–100 nm) has suggested geometrical rules governing outer coat assembly, and indicated regions of

**eLife digest** Proteins often need to move between different compartments within cells. To do this they are packaged into transport pods called vesicles. Many trafficked proteins are synthesized in an organelle called the endoplasmic reticulum, or ER; these proteins are transported away from the ER in 'COPII' vesicles, which are formed when the COPII proteins assemble on the ER membrane and force it to bulge outward. The bulge pinches off from the ER membrane, forming the vesicle, which can then move to, and fuse with, a different compartment in the cell.

The COPII proteins assemble in a particular order to form the vesicle—Sar1 inserts into the membrane of the ER; Sec23 and Sec24 form an inner coat and capture the proteins that the vesicle will transport; and Sec13 and Sec31 form an outer coat. Although the structures of the coat proteins are known, how they bind to each other—and to the ER membrane—to form vesicles of many shapes and sizes is less well understood. Now, Zanetti et al. show how the inner and outer coat proteins can interact flexibly to accommodate a variety of cargoes.

Zanetti et al. mixed purified Sar1 and COPII coat proteins with artificial membranes in vitro. As in cells, the proteins assembled a coat on the membranes, creating bulges and vesicles of different shapes. These coats were imaged using an electron microscope, and the images were analysed using computational image-analysis methods. In this way, Zanetti et al. produced a detailed 3D view of the assembled coat.

It was found that the inner and outer proteins each arranged to form lattice structures—like fishing nets—which showed flexibility and variability in the way the individual proteins interact, as well as imperfections in the arrangement. Both coats may help to reshape the membrane, and the inner-coat and outer-coat lattices were also found to move with respect to each other. These flexible properties could allow the coat to assemble on membranes with different shapes and curvatures, forming COPII vesicles with distinct sizes and shapes that can carry a range of cargoes.

flexibility in Sec13/31 that permit envelopment of vesicles with sizes ranging from 60 to 120 nm (**Fath et al., 2007**; **Stagg et al., 2008**; **Bhattacharya et al., 2012**). Nevertheless, a higher degree of flexibility in COPII architecture is needed to explain the ability of COPII to mediate secretion of cargoes such as 300 nm pro-collagen fibres, which are much larger than the 60–100 nm in vitro assembled cages. There is increasing evidence that incorporation of pro-collagen into COPII coated vesicles is a highly regulated process, and that modifications in the outer coat proteins—such as ubiquitination—as well as in timing of GTP hydrolysis and coat recycling may be necessary for COPII ability to coat large vesicles (**Saito et al., 2009**; **Jin et al., 2012**; **Kim et al., 2012**; **Venditti et al., 2012**). COPII-dependent defects in collagen transport are linked to a genetic syndrome characterized by late-closing fontanels, sutural cataracts, facial dysmorphisms and skeletal defects: Cranio-lenticulo-sutural dysplasia (**Boyadjiev et al., 2006**).

There are currently no structural data on the COPII coat assembled in its functional state on a membrane, leaving important mechanistic questions unanswered. How do the inner and outer coat components arrange to form a membrane coat? How do they interact with the membrane? How do they interact with each other? Are the existing models for the outer COPII coat cages representative of the complete coat assembled on a membrane? What is the structural basis for COPII ability to transport large or elongated cargoes such as pro-collagen? Here we address these questions. We provide evidence for coordinated assembly of the two layers of the coat on a lipid membrane and suggest how this interplay may effect shape changes essential to the capture of large and unusually-shaped secretory cargo complexes and particles.

## Results and discussion

To answer the questions above, it is necessary to understand the structural arrangement of the complete coat in its membrane bound form. To this end we have studied a reconstituted COPII assembly reaction (**Bacia et al., 2011**). We incubated giant unilamellar vesicles (GUVs) with yeast COPII proteins (Sec12, Sar1, Sec23/24, Sec13/31) in the presence of a non-hydrolysable GTP analogue GMP-PNP. We imaged the reaction mixture by cryo-electron tomography, observing that coated membranes of a variety of shapes and sizes were generated. These included round vesicles, either independent or still

attached to the donor membrane, as well as curved or extended tubes, reminiscent of those detected budding from ER exit sites in vivo (*Mironov et al., 2003*; *Fromme et al., 2007*; *Venditti et al., 2012*; *Figure 1*). All of these structures were surrounded by two protein layers indicating that the COPII coat can assemble on membranes with a range of different curvatures. The outer coat appeared in sections as points or lines (*Figure 1A*), and in grazing slices as a regular arrangement of X-shaped features arrayed to form a rhomboidal lattice (a lozenge pattern) (*Figure 1B,C*). Other geometries, such as triangles and pentagons, were observed on vesicles and tube ends (*Figure 1D*). These geometries are similar to those formed when purified Sec13/31 proteins assemble in vitro in the absence of a membrane (*Stagg et al., 2006*, *2008*). The inner coat layer also appeared as a regular array (*Figure 1B*), implying that Sar1 and Sec23/24 can also assemble to form a lattice.

## The structure of the outer coat

To understand the architecture of the Sec13/31 outer coat we applied reference free, contrast-transfer function (CTF) corrected, subtomogram averaging to solve the 3D structure of the vertex, and of the connecting rods to resolutions of ~40 Å (*Figure 2A–B*, *Figure 2—figure supplement 1*, and 'Materials and methods'). The vertex structure was a twofold symmetric X-shape, similar to that seen in in vitro assembled Sec13/31 cages (*Stagg et al., 2006*, *2008*) (*Figure 2D*, *Figure 2—figure supplement 2*). The connecting rods are consistent in shape and size with Sec13/31 heterotetramers, and are bent in the middle by approximately 15°. This same bend is seen in the solved X-ray crystal structure, which can be fitted into the densities as a rigid body (*Figure 2B,C*). In contrast, there is a 45° bend in the rods of in vitro-assembled protein cages (*Stagg et al., 2008*; *Figure 2E,F*). These data indicate that the central hinge between Sec31 molecules can adapt to assemble coats of different curvatures.

The published EM reconstructions of in vitro assembled COPII outer coat protein cages reveal that the Sec31 β-propeller domains at the ends of two Sec13/31 rods (referred to in the literature as the plus ends) contact each other directly at the center of the vertex, whereas the ends of the other two rods (the minus ends) contact the plus ends at the side and are more distant from the center of the vertex. The geometrical relationship of the '+' and '−' rod ends at the vertex can be described by two

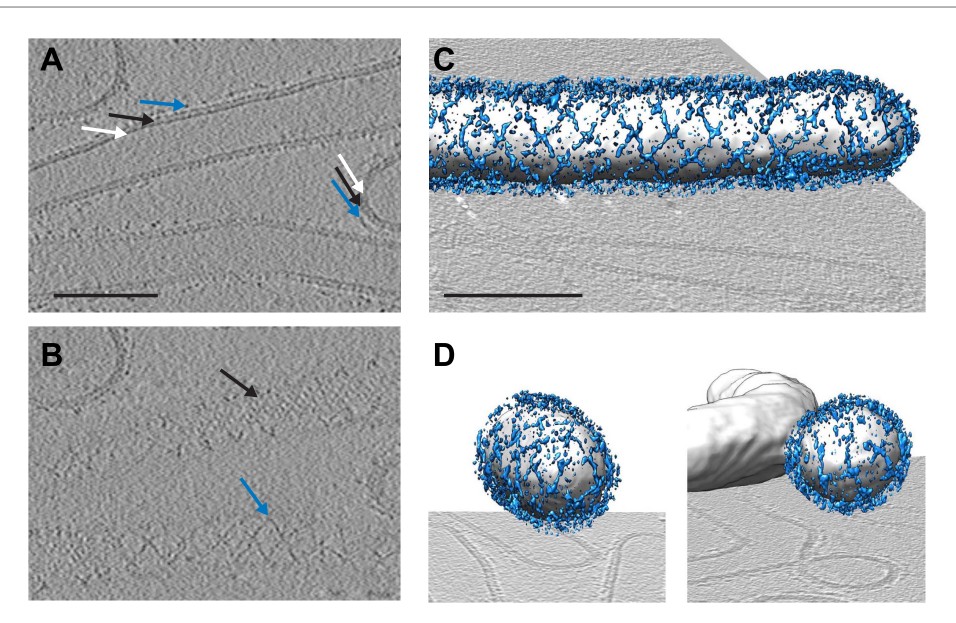

**Figure 1**. Cryo-electron tomograms of a reconstituted COPII budding reaction. Scale bars = 100 nm. (**A**) A slice through a tomogram showing two coat layers arranged around tubular and spherical membranes. White, black and blue arrows point to the membrane, inner, and outer coat layers respectively. (**B**) A slice through the top of the tubes in panel **A**, showing repeating features in the coat layers. (**C**) A surface rendering of a COPII-coated tube. The membrane and inner coat are in grey, the outer coat in blue. (**D**) Surface renderings of spherically curved regions of membrane, coloured as in panel **C**.

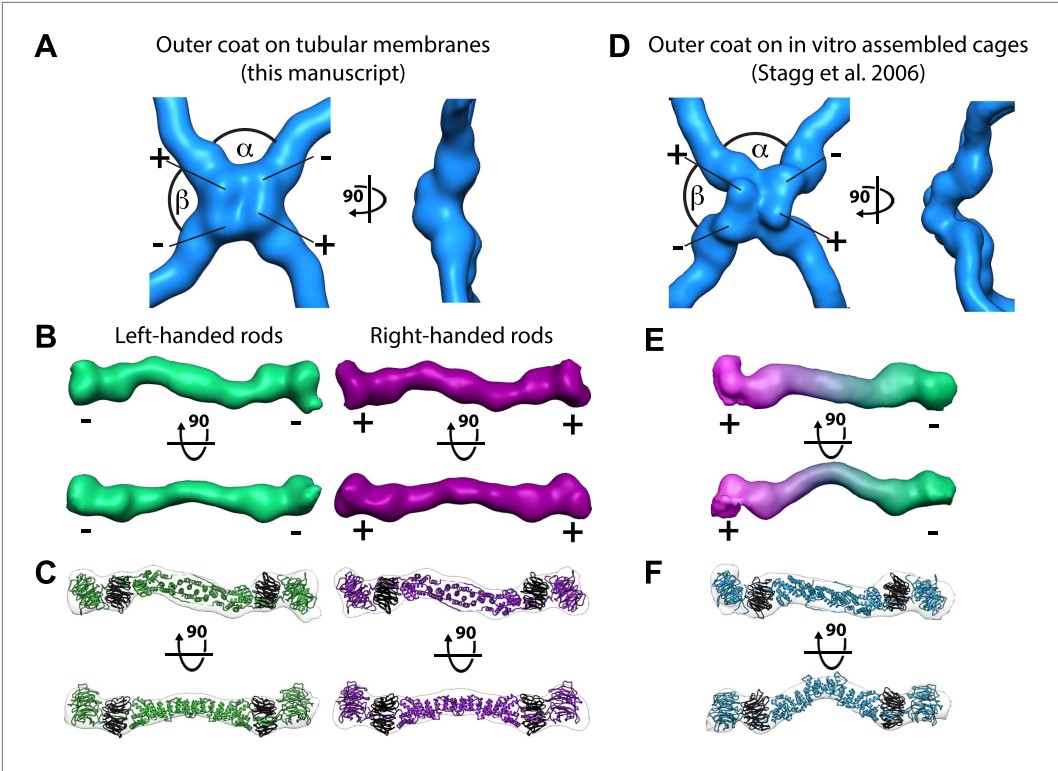

**Figure 2**. Structure of the outer COPII coat. (**A**) Isosurface representation of the outer coat vertex solved by sub-tomogram averaging of tubular membranes. '+' and '−' ends of Sec13/31 and alpha and beta angles are as defined by (*Stagg et al., 2008*) (*Figure 2—figure supplement 2* and panel **D**). (**B**) Structures of the rods that interconnect neighbouring vertices in left-handed (green) and right-handed (purple) helical directions, viewed from the top (upper panels), and the side (lower panels). Left-handed rods have two '−' ends, whereas right-handed rods have two '+' ends. (**C**) Atomic model of the Sec13/31 complex (*Fath et al., 2007*) (PDB 2PM9 and 2PM6) fitted as a rigid body into left- and right-handed rods. (**D**) Isosurface representation of the Sec13/31 vertex structure from cryo-electron microscopy of in vitro assembled cuboctahedral cages (*Stagg et al., 2006*) (EMDB ID 1232). (**E**) Structure of the rods segmented from in vitro assembled cuboctahedral cages (*Stagg et al., 2006*) (EMDB ID 1232) for comparison. '+' and '−' ends are coloured purple and green respectively. (**F**) The Sec13/31 complex fitted into a rod from the cuboctahedral cage. To adapt to the 45° bend in the rod two equivalent heterodimers were fitted independently.

The following figure supplements are available for figure 2:

**Figure supplement 1**. Resolution of outer coat structures.

**Figure supplement 2**. Previously published outer coat structures.

angles: alpha (clockwise between + and − ends, which is 60° in the cages), and beta (clockwise between − and + ends which is variable at least between 90° and 108° in the cages) (*Figure 2A,B*, *Figure 2— figure supplement 2*, *Figure 3—figure supplement 1*) (*Stagg et al., 2008*). Within the in vitro assembled cages each Sec13/31 rod makes a '+' contact at one end and a '−' contact at the other end. This arrangement allows assembly of a coat that curves in two directions, appropriate for coating spherical membranes. (*Figure 3C*, *Figure 3—figure supplement 1*). In our reconstructions, the arrangement of Sec13/31 rods we observed at the rounded tips of tubes and on spherical membranes (*Figure 1D*) was similar to that seen in in vitro assembled cages, consistent with the presence of such '+/−' rods (*Figure 3C*). When we analysed the distribution of vertices (*Figure 3A*) and rods (*Figure 3B*) on the tubular membranes we found that they assembled regions of rhomboidal lattice. A rhomboidal lattice could be built in two ways: each vertex could be rotated by approximately 90° in the plane of the membrane relative to the adjacent vertices, or each vertex could have the same orientation (see

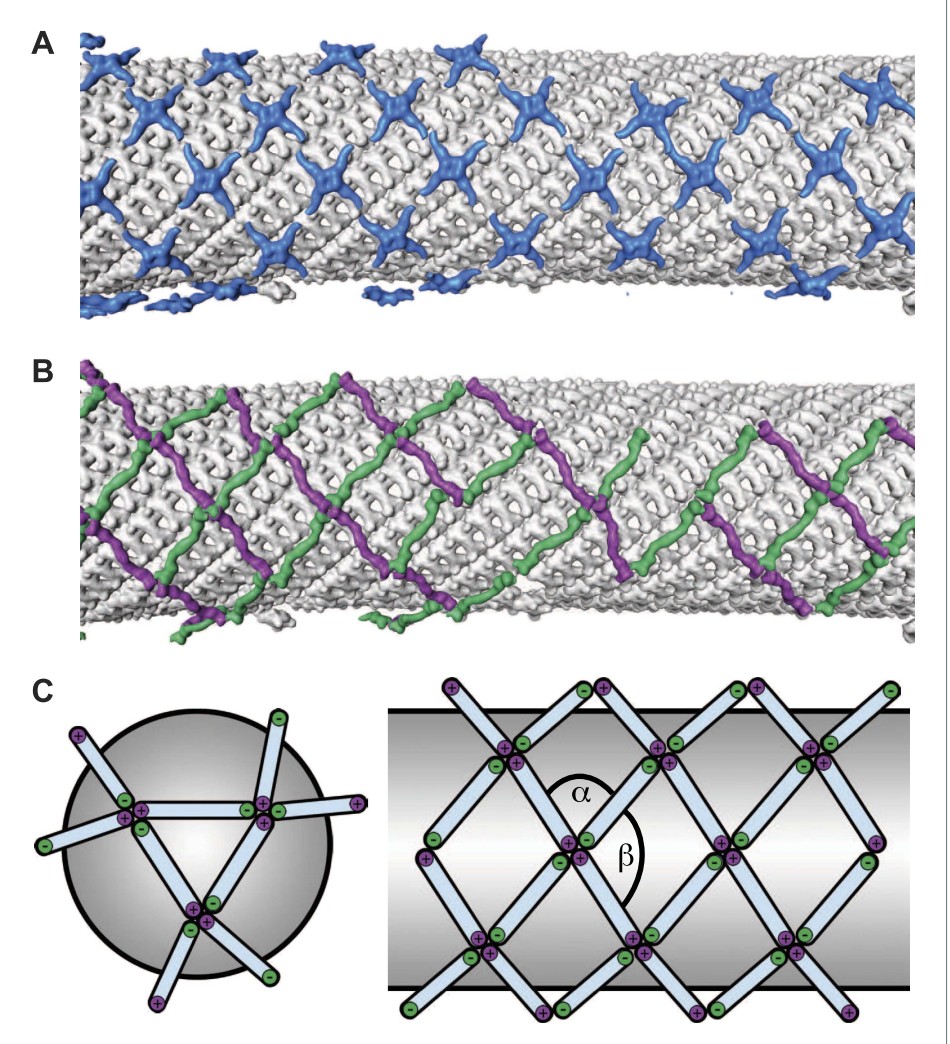

**Figure 3**. Arrangement of the outer COPII coat. (**A**) Visualization of the positions and orientations at which vertices were identified for a representative tube. They are arranged to form a rhomboidal lattice. The positions of inner coat subunits are shown in grey (*Figure 4*). (**B**) Visualization of the positions of aligned rods, as in panel **A**. Right- and left-handed rods in purple and green respectively. (**C**) Schematic depiction of how '+/−' rods can coat regions of spherical curvature by arranging to form orthogonal vertices (left panel). Tubular surfaces (right panel) are coated with +/+ and −/− rods that form parallel vertices (alpha and beta are always in the same direction with respect to the tube axis).

The following figure supplements are available for figure 3:

**Figure supplement 1**. Current models for assembly and size variation of outer COPII coat cages.

**Figure supplement 2**. Variability and flexibility in the outer coat calculated from the coordinates of aligned subtomograms.

**Figure supplement 3**. The unit cell dimensions for the inner and outer coat lattices.

---

schematic in *Figure 3—figure supplement 2C*). The final structure of the vertex on the tubes clearly shows the expected twofold features of the vertex previously described by Stagg et al. (*Noble et al., 2012*), indicating that the majority of vertices have the same orientation ('Materials and methods'). In this arrangement of vertices the rods oriented in one direction (the right handed rods, purple in *Figures 2C and 3B*) each make two '+' contacts (+/+), while the left-handed rods (green in *Figures 2C*

*and 3B*) each make two '−' contacts (−/−). The alpha angle is 79.7° ± 5.9° and is oriented around the tube circumference while the beta angle is 95.7° ± 5.8° and is oriented along the tube axis (*Figure 3—figure supplements 2 and 3*). Together these data imply that three properties contribute to outer coat adaptability: (i) variability of both alpha and beta angles at the vertices, (ii) flexibility of the central rod hinge, and (iii) the absence of any inherent asymmetry in the Sec13/31 rods, allowing them to make '+' contacts at both ends, '−' contacts at both ends, or a '+' contact at one end and a '−' contact at the other. Together this versatility allows coating of not only spherical, but also of tubular membranes and therefore accommodation of large elongated cargoes such as pro-collagen.

Proteins able to form both spherical and tubular structures are found in other biological systems, most notably virus capsids. For example, the elongated heads of T-even phages, as well as the capsid cores of retroviruses, are assembled as closed structures built from hexamers and pentamers of the component protein and can have regions with spherical and with tubular curvatures (*Baschong et al., 1988*; *Ganser et al., 1999*). This structural flexibility can be understood within the framework of quasi-equivalence: while subunits are found in symmetrically different positions with different local curvatures, the contacts between the subunits are only subtly different (*Caspar and Klug, 1962*). In some virus capsids, subtle changes in the contacts are supplemented by structural switches that mediate the change from hexameric to pentameric assembly (*Johnson and Speir, 1997*). It seems likely that flexibility in the outer COPII cage geometry is primarily mediated by subtly varying contacts at multiple hinge positions in the rod and vertex (*Noble et al., 2012*), but we cannot rule out the presence of structural switches.

## The structure of the inner coat

Our images showed that the inner coat can also form a regular lattice, suggesting it may not only function to link cargo and membrane to the outer coat, but also play a structural role in determining vesicle shape. We applied reference-free, CTF corrected, subtomogram averaging to solve the structure of the repeating inner coat unit with a resolution of approximately 23 Å (*Figure 4A*, *Figure 4—figure supplement 1*; 'Materials and methods' and *Faini et al., 2012*). The bow-tie-shaped repeating unit interacts tightly with the outer membrane leaflet. The units interact with each other to form an approximately helical array on the tube surface (*Figure 4B*). In many instances we see step changes in tube diameter, associated with displacement of a few subunits from the lattice (*Figure 4B*).

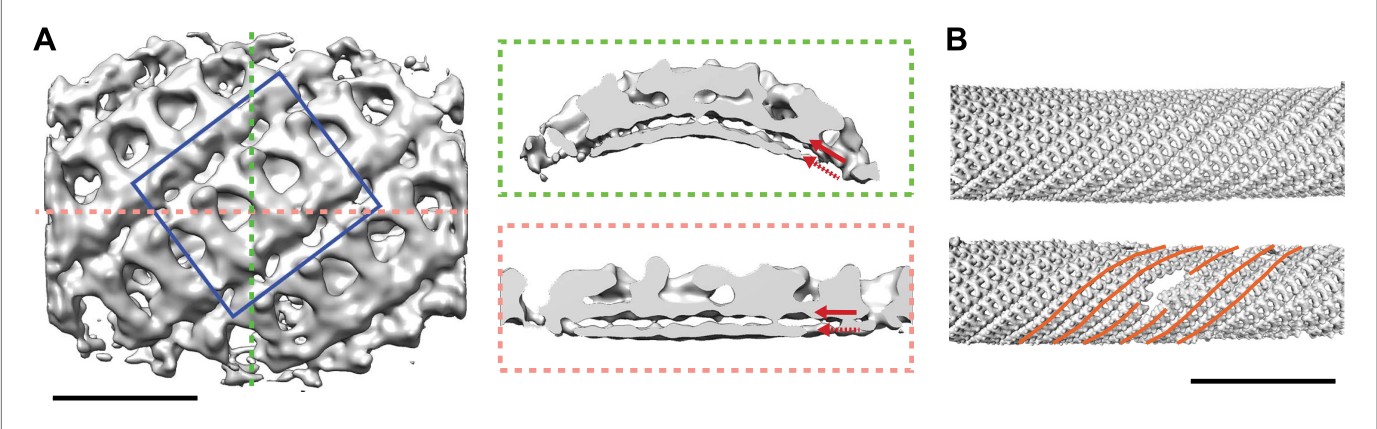

**Figure 4**. Structure and arrangement of the inner COPII coat. (**A**) Isosurface representation of the inner coat on tubular membranes solved by sub-tomogram averaging. Top view and perpendicular slices cut along the dotted lines (green and pink boxes) are shown. Inner and outer leaflets of the lipid bilayer are indicated by dotted and continuous red arrows, respectively. Scale bar 20 nm. (**B**) Inner coat subunits placed in their aligned positions and orientations on two representative tubes. At positions where tube diameter changes, the helical symmetry changes (orange lines). Scale bar 100 nm.

The following figure supplements are available for figure 4:

**Figure supplement 1**. Resolution of inner coat structure.

We docked the crystal structures of Sec23/Sar1 (PDB 2QTV), and of Sec23/24 (PDB 1M2V) (*Bi et al., 2002*) into the subtomogram averaging electron density map. Both fit unambiguously into one bow-tie shaped subunit as rigid bodies, in each case placing Sec23 in the same position ('Materials and methods'; *Figure 5A*). Analysis of the fitted models reveals the novel network of interactions that mediate lattice formation and membrane binding. Two interfaces between the long edges of the Sar1/Sec23/24 heterotrimer mediate formation of lattice rows. The larger involves Sar1, the gelsolin-like and the C-terminal domain of Sec23 from one heterotrimer interacting with the Zn-finger and part of the beta barrel domains of Sec23 in the adjacent heterotrimer. The smaller interface is between the Zn-finger domain of Sec24 in one heterotrimer and the gelsolin-like and C-terminal domains of Sec24 in the neighbouring heterotrimer (*Figure 5B*). These contacts are characterised by extended surfaces with opposite charges, a property that is conserved (*Figure 5—figure supplement 1*). The rows contact each other where the beta barrel domain of Sec23 approaches the Sec24 N-terminus. In the Sec23/24 crystal structure (PDB ID 1M2V), the Sec24 N-terminal helix 61-73 forms a crystallographic contact with the beta-barrel domain of Sec23 in the neighbouring asymmetric unit (*Bi et al., 2002*). When placed in this position, the 61–73 helix fills an unoccupied region of electron density in our structure, suggesting this contact is present in the assembled coat (*Figure 5A,B*, and 'Materials and methods').

Our structure reveals that the concave surface of the inner coat heterotrimer is oriented towards the membrane and the Sec31 binding site towards the coat exterior (*Figure 5A,C*), as previously predicted (*Bi et al., 2002*). We identify two primary membrane interacting regions. One is formed by a small part of Sec23 together with Sar1, orienting Sar1 to direct its N-terminal amphipathic helix into the outer leaflet. The other is a large basic surface in Sec24, including the Sec24 Zn finger, beta barrel and alpha helical domains (*Figure 5B*). While the Sec24 Zn finger domain binds the membrane, the Sec23 Zn finger domain does not, instead it faces the neighbouring heterotrimer, suggesting that the role of the Zn fingers in Sec23 and Sec24 has diverged. Consistent with this difference in function, the Sec24 Zn finger, essential for cell survival (*Peng et al., 1999*), contains a conserved basic patch that is absent in Sec23. We speculate that, analogous to the basic Zn fingers in FYVE domains (*Gaullier et al., 2000*), this represents a phospholipid-binding site. The trunk domains of Sec23 and Sec24 do not contact the membrane but are suspended above it, leaving space that would allow binding to more highly curved vesicles (*Figure 5B*).

Inner coat assembly into lattices must remain compatible with the well-characterised cargo binding activity of Sec24. To test this, we modelled the X-ray structures of Sec24 co-crystallised with cargo-derived peptides (*Mossessova et al., 2003*; *Mancias and Goldberg, 2007*, *2008*), or the cytosolic domain of a globular cargo Sec22 (*Mancias and Goldberg, 2007*) in our map. None of the characterised cargo-binding sites overlap with the protein–protein and protein–membrane interfaces we identified in the lattice: all remain accessible to cargo in the assembled inner coat (*Figure 5C,D*). Strikingly, comparison of the co-crystal structure of Sec23/24/Sec22 with our density map reveals that the space between heterotrimers in the assembled inner coat lattice forms a pocket into which Sec22 would bind (*Figure 5C*).

We cannot rule out that the large-scale order seen in the inner coat lattice is promoted by carrying out assembly in vitro in the absence of GTP hydrolysis. Three factors suggest, however, that the interactions we observe between heterotrimers are relevant in vivo and are not simply an artefact of in vitro assembly. Firstly, they result from a productive assembly reaction that bends the GUV membrane into a variety of shapes. Secondly, the interfaces have conserved electrostatic properties. Thirdly, the interactions that mediate inner coat lattice assembly are distinct from those that mediate membrane binding, outer coat binding, and cargo binding and can accommodate Sec22 in the correct position: they are fully compatible with available biochemical and structural data.

## The relationship between inner and outer coats, and implications for assembly

The low resolution that can be obtained in cryo-EM reconstructions of Sec23/24 bound to in vitro assembled Sec13/31 cages, and the necessity of implying symmetry which may not be appropriate for both coat layers, has prevented a clear understanding of the structural relationship between the inner and outer coats (*Stagg et al., 2008*; *Bhattacharya et al., 2012*). Our structure of the outer coat does not reveal density corresponding to an ordered inner coat, and vice versa, indicating that the relative positions of the two coat layers are not fixed. Nevertheless, the two layers are approximately aligned

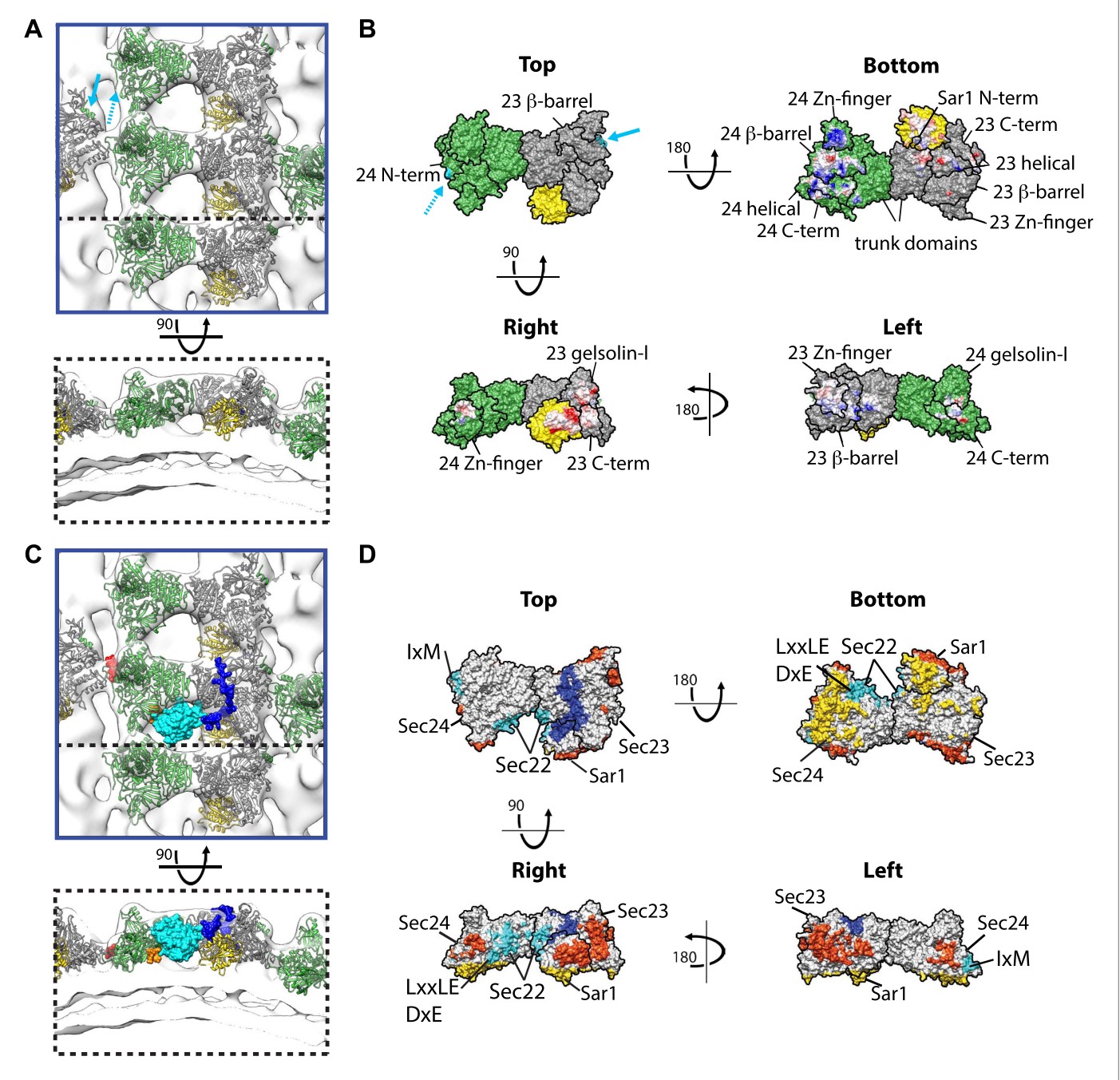

**Figure 5**. Interactions within inner coat lattice and membrane. (**A**) Magnified view of the blue-outlined region in *Figure 4A*, viewed from top (upper panel) and side (lower panel). Atomic model of the inner coat (PDB 1M2V and 2QTV) is fitted into the reconstruction (Sec24 green, Sec23 grey, Sar1 yellow). Helix 61–73 of Sec24 (solid cyan arrow) is shown in the position where it binds Sec23 in the Sec23-24 co-crystal structure. Residue 133 of Sec24 is marked (dotted cyan arrow) ('Materials and methods'). (**B**) Surface representations of the fitted atomic model with boundaries between domains marked. Cyan arrows as in **A**. Residues interacting with the membrane ('bottom view') or mediating protein-protein interactions ('right' and 'left' views) are coloured based on coulombic potential (red: negative, blue: positive). (**C**) Atomic models of Sec23/24 co-crystallised with cargo or Sec31 active fragment superimposed to the fitted model: the IxM containing peptide from syntaxin 5 (PDB 3EFO) (red), the Sec22 cytosolic domain (PDB 3EGD) (cyan), a DxE containing peptide derived from VSV-G (PDB 3EGD) (orange), and the Sec31 active peptide (PDB 2QTV) (blue). (**D**) As in **B**, with surfaces coloured by interaction partner: membrane (yellow), cargo (cyan), Sec31 (blue), and neighbouring Sar1/Sec23/24 heterotrimers in the lattice (orange).

The following figure supplements are available for figure 5:

**Figure supplement 1**. Interactions between inner coat subunits are mediated by surfaces with conserved electrostatic properties.

both in spacing and orientation. The 'rows' in the inner coat are aligned to the left-handed rods of the outer coat, with four inner coat heterotrimers spanning one outer coat rod. The inner coat 'columns' are aligned to the right-handed outer coat rods, with two inner coat heterotrimers spanning one outer coat rod (*Figure 3A,B*, *Figure 3—figure supplement 3*). This relationship suggests that the unstructured C-terminal region of Sec31 (*Fath et al., 2007*; *Noble et al., 2012*), which connects the inner and outer coats, constrains the coat layers but does not fix their absolute positions, much like multiple anchor cables on a ship. This flexibility in the relative positions of the two layers would permit adaptation of the coat to variable curvatures. Gaps and dislocations in both inner and outer coat lattices may further contribute to curvature variability.

The spherical tube ends have an outer coat arrangement similar to that previously observed in in vitro cages, in which adjacent vertices are rotated relative to one another. A helical array of the inner coat heterotrimer cannot be present on the spherically-curved tips. Instead, we consider it likely that on the tube tips, the inner coat lattice is arranged in smaller patches rotated relative to one another, each maintaining the same relationship to the overlying outer coat vertex. Gaps between or within lattice patches could accommodate cargoes with large membrane-proximal domains. A comparison of our results with previous hypotheses for the relationship between inner and outer coats (*Stagg et al., 2008*; *Bhattacharya et al., 2012*) is in *Figure 6*.

The structural data we have presented shows that both inner and outer COPII layers have sufficient structural versatility to coat both spherical and tubular membranes, and thereby accommodate important large cargoes such as pro-collagen. We found that both coats can assemble regular lattices

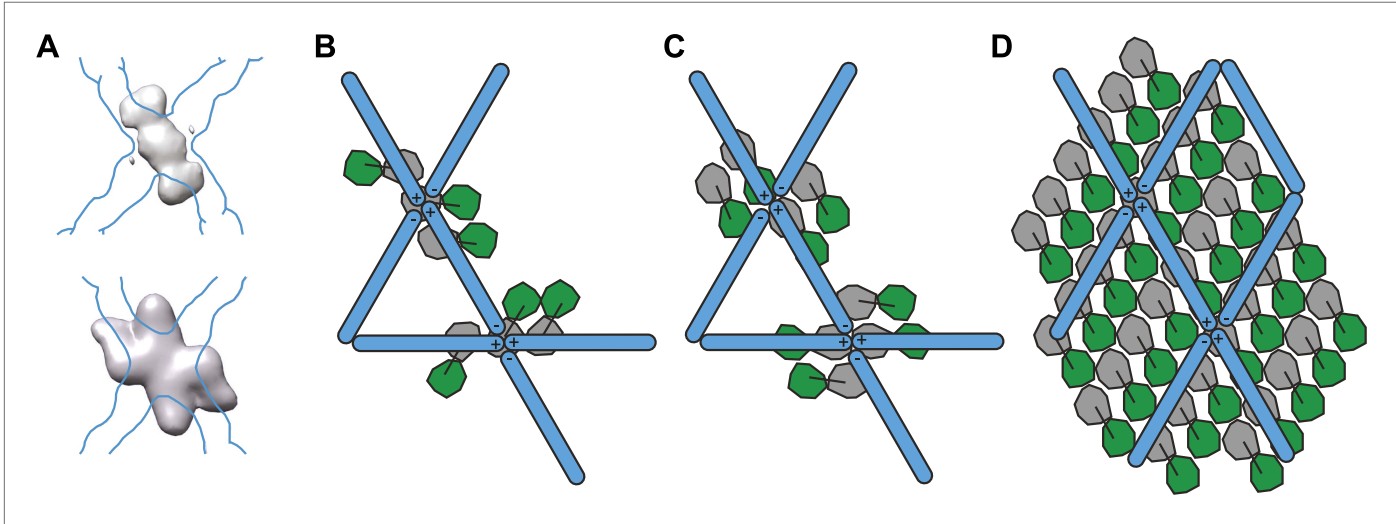

**Figure 6**. Models for the spatial relationship between inner and outer coat. (**A**) Cryo-electron microscopy reconstructions of COPII cages assembled in vitro in the presence of inner coat subunits, but in the absence of a membrane, showed additional density below that of the outer coat. The top panel shows the density for the inner coat obtained in cages formed in the presence of Sec23 alone (*Bhattacharya et al., 2012*). The bottom panel shows the density for the inner layer in cages formed in the presence of Sec23/24 (*Stagg et al., 2008*). The position of the outer coat is marked as a blue outline. Note that in both studies, the twofold symmetry of the outer coat vertex was imposed on the inner coat. (**B**) Based on the comparison between the structures in panel **A**, Stagg et al. proposed that Sec23 binds at two positions: one copy below each vertex and two copies at its side. Sec24 would be located under the holes in the cage (*Bhattacharya et al., 2012*). According to this model, the inner coat arrangement does not conform to the twofold symmetry applied during the reconstruction procedure, explaining why it is poorly resolved in the cage reconstruction. (**C**) A previous model was proposed based on the reconstruction of icosidodecahedral cages (bottom panel in **A**). In this model, Sec23/24 subunits are oriented with their long axes in the direction of the +/+ connection of the outer coat vertex. Two arrangements consistent with this model are shown: in the first (top) the subunits all have the same orientation, in the second (bottom), subunits have opposite orientations, following the twofold symmetry of the outer coat. (**D**) A schematic representation of the relationship between inner and outer coat as identifed in this study. This arrangement is similar to the model suggested in **C** (top). The stoichiometry of the two coat layers is 1:2 outer:inner in coated tubes, meaning that only half of the inner coat subunits are bound by the known Sec31 interface. The percentage of bound inner coat subunits would be higher on spherical vesicles. The difficulty in resolving the position of the inner coat in the studies of cages assembled in the absence of a membrane can be explained by our observation that the relative positions of outer and inner coats are constrained but not fixed, and that the two layers do not follow the same symmetry.

with constrained relative positions. This relationship would permit the inner coat to influence the arrangement of the outer coat and therefore to play a critical structural role in determining vesicle shape. Our data hint at the possibility that the formation of larger arrays of the inner coat and the formation of associated rhomboidal outer coat lattices, could favour the formation of larger tubular carriers able to accommodate cargoes such as pro-collagen. It has previously been proposed that proteins such as TANGO1 and Sedlin (*Saito et al., 2009*; *Jin et al., 2012*; *Kim et al., 2012*; *Venditti et al., 2012*) act in concert to stabilize the inner coat, modulating Sar1 GTPase activity and delaying both outer coat recruitment and scission. We suggest that stabilization of the inner coat could promote growth of larger carriers not only by delaying scission, but also by promoting tubular membrane morphology (*Figure 7*).

## Materials and methods

### Sample preparation

Expression and purification of yeast COPII proteins, as well as GUV production by electroformation, were performed essentially as described in *Bacia et al. (2011)*. In vitro reconstitution of coat assembly was performed by adding Sar1p (2 µM), Sec12ΔCp (1 µM) and GMP-PNP (1 mM), Sec23/24p (320 nM) and Sec13/31p (520 nM) to 2 µl of GUVs in 20 mM HEPES, pH 6.8, 50 mM KOAc, 1.2 mM MgCl$_2$ (final volume 40 µl). After incubation at room temperature for 2 hr the undiluted reaction was mixed with 3 µl Protein-A conjugated 10 nm colloidal gold as fiducial markers for tomography, applied to glow discharged C-flat (Protochips Inc.) holey carbon coated grids and vitrified by plunge freezing.

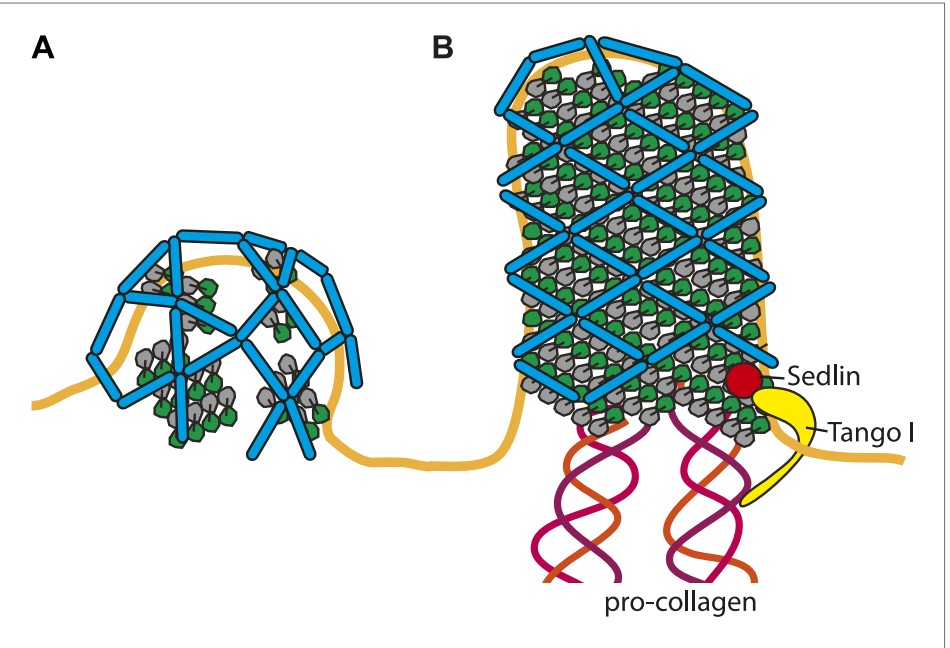

**Figure 7**. Cartoon depiction of a model for two COPII assembly modes. (**A**) On a spherically curved membrane the bow-tie shaped inner coat subunits assemble in small patches that may be randomly oriented with respect to each other. Outer coat rods bind to the inner coat patches in a preferred orientation. The outer coat can assemble to form triangles, squares, or pentamers. (**B**) If the inner coat forms large arrays instead of small patches, then the outer coat, interacting with the inner coat in its preferred orientation, will tend to arrange to form a lozenge pattern. This arrangement of inner and outer coats results in coated tubular membranes. This arrangement could simply be promoted by packaging of elongated cargoes. It could also be favoured when external factors intervene to either delay outer coat recruitment, and/or stabilise larger inner coat patches. Tango I and Sedlin (*Saito et al., 2009*; *Venditti et al., 2012*) have been proposed to facilitate formation of large COPII carriers that are capable of incorporating 300 nm-long procollagen molecules, and to achieve this by stabilizing the inner coat on the membrane.

## EM

CET was performed on FEI Titan Krios electron microscope operated at 200 kV at liquid nitrogen temperature equipped with a Gatan GIF 2002 post column energy filter and a 2k × 2k Multiscan charge-coupled device camera. Tilt series were collected with an angular range of −60° to +60°, angular increment of 3°, defocus 1.5–6 µm, total electron dose 80 e$^-$/Å$^2$ and a magnification of 19,500×, giving a pixel size of 4.3 Å at the specimen level.

## Data processing

Tomograms were aligned based on the positions of gold fiducials and reconstructed by weighted back-projection, using the IMOD (*Kremer et al., 1996*) and raptor (*Amat et al., 2008*) software packages. Visual analysis and segmentation of reconstructed tomograms were performed with Amira (Visualisation Sciences Group, an FEI company) and Chimera (*Pettersen et al., 2004*). Sub-tomogram averaging was carried out essentially as described previously (*Faini et al., 2012*) using Matlab scripts adapted from the TOM/AV3 tomography toolbox (*Förster et al., 2005*) and using the Dynamo package (*Castano-Diez et al., 2012*).

Two datasets collected on the FEI Titan Krios were used for sub-tomogram averaging. A 'far from focus' dataset collected at nominal defoci of 4–6 µm contained 16 tubes (typically 1 tube in each tomogram). A second, close to focus dataset (nominal defoci 1.5–2.5 µm), contained 26 tubes. Both unbinned datasets had a pixel size of 4.3 Å.

### Inner coat sub-tomogram averaging

Subtomograms were extracted in 128$^3$ boxes at uniformly distributed points on the surfaces of tubes, spaced by 15 pixels, and initial Euler angles were assigned based only on the position of the subtomogram relative to the tube axis. Initially, subtomograms from the far-from-focus dataset were averaged without alignment to generate a smooth starting reference, and then iteratively aligned against the reference using an adapted version of the TOM/AV3 package, as previously described (*Faini et al., 2012*). The final average of the complete dataset (reference A) showed a helical lattice formed by twofold symmetric subunits, but averages of subtomograms from individual tubes showed that each tube has a defined directionality. Subtomograms from the close to focus dataset were therefore aligned against reference A, and individual tube averages were observed to determine the tube directionality.

The close-to-focus dataset was CTF corrected (see section 'CTF determination and correction' for details), and subtomograms were extracted in 128 boxes and binned to a pixel size of 8.6 Å. The over-sampled dataset contained ~90,000 subtomograms, and was divided into two halves with one half of each tube in each half dataset. In order to control for reference bias and overalignment, each subtomogram was assigned starting Euler angles based only on the position of the subtomogram relative to the directional tube axis. The two halves of the dataset were then subjected to independent reference-free, iterative alignment with a low-pass filter set at 45 Å. To increase speed, only one in 10 subtomograms of each half dataset was initially included. The independent final averages were unbinned, shifted and oriented to place the subunits in the same position, and cropped. They were then used as references for independent alignments of the two full datasets of unbinned subtomograms extracted into 64$^3$ boxes. After one round of alignment, 'duplicate' subtomograms that had aligned to the same position as another subtomogram were removed, and clearly misaligned particles (those not oriented to the surface of the tube) were excluded by setting a cross correlation threshold for each individual tube. Seven further refinement iterations generated two convergent averages. Resolution was estimated by Fourier shell correlation between the two independent averages after masking with a spherical mask with a 5 pixel Gaussian fall-off. The complete dataset was then combined into the final reconstruction which contained a total of ~15,000 subtomograms.

The structure was further subjected to amplitude weighting (considering the weighted contribution of CTF amplitudes and assuming random orientation of all sub-tomograms) and sharpening (B = 2500, multiplied by the FSC between the two half datasets) to yield the final electron density map presented here.

### CTF determination and correction

Averages aligned against reference A were individually refined for each tomogram. The CTF parameters for each tomogram were estimated by calculating averaged structures from all individual tomograms,

measuring all the pairwise FSCs between these averages, and observing frequencies of positive and negative correlation. Averaged structures from distinct tomograms will correlate positively until the mean first zero of the further from focus tomogram, the correlation will then be negative until the closer to focus tomogram reaches its mean first zero, or the further from focus tomogram reaches its second zero. The FSCs between the average structure from one single tomogram with the structures obtained from every other individual tomogram should therefore all intersect at the first zero of the single tomogram. Using this estimate of defocus, CTF correction was applied to each image as described in (*Zanetti et al., 2009*) taking tilt into account. CTF parameters were then refined using the FSC between individual uncorrected averages and the global structure obtained after initial CTF correction (these FSCs have an inversion at the first zero of the tomogram in question). The measured defoci in the close to focus dataset ranged from 2.0–3.25 μm. An approximate CTF correction is sufficient to recover data up to the frequencies of interest (*Zanetti et al., 2009*). A validation of the CTF correction is shown in *Figure 4—figure supplement 1*.

## Outer coat sub-tomogram averaging

The positions of outer coat vertices were manually marked and 128 × 128 × 128 pixel subtomograms were extracted from CTF-corrected tomograms at these positions. Initial Euler angles were assigned based on the angles of the closest aligned inner coat subunit, and the subtomograms were iteratively aligned against their total sum, using a 40 Å lowpass filters at each iteration. The inner coat was masked out during the alignments. Alignments were carried out using AV3 (*Förster et al., 2005*), and refined using Dynamo (*Castano-Diez et al., 2012*). After six rounds of alignment we confirmed that the vertices are twofold symmetric, as seen for in vitro assembled cages. Particles that were obviously misaligned (i.e., too far from the coat layer surface or not laying flat on the tubes), were removed based on visual inspection of placed sub-tomograms within the original tomogram, and six alignments iterations were carried out on a 417 sub-tomogram dataset, applying twofold symmetry. The final structure was weighted for CTF amplitudes, and sharpened (B = 6000). An indicative resolution of 40 Å was assessed, judging from the FSC between two 180° rotated non-symmetrised copies of the final structure using the 0.5 threshold criterion.

The coordinates of each pair of neighbouring vertices were used to estimate the distance between vertices in the right-handed and left-handed directions, as well as the angles between triplets of vertices (*Figure 3—figure supplements 2 and 3*).

The datasets for right- and left-handed rods were extracted for sub-tomogram alignment at positions equidistant between aligned vertices. Initial Euler angles were assigned to each rod based on its position relative to the directional tube axis. Sub-tomograms were iteratively aligned against their total sum. A total of 182 sub-tomograms contributed to the structure of left-handed rods, and 192 sub-tomograms to that of the right-handed rods. The final structures were subjected to CTF-amplitude weighting and sharpening (B = 6000). The FSCs were calculated between the structures and their 180° rotated version.

## Atomic docking

Automatic fitting of inner COPII units was done using Chimera (*Pettersen et al., 2004*). Independent fitting of the X-ray atomic models Sec23/Sar1 (PDB 2QTV) and Sec23/24 (PDB 1M2V) in the central segmented unit placed Sec23 in the same position, with an RMSD between the two Sec23 molecules of about 3 Å. We therefore created a single PDB file containing Sar1, Sec23 and Sec24 and fitted it as a rigid body. The N-terminus of Sec24 is absent from the X-ray model. Only helix 61–73 is resolved, detached from the rest of the sec24 molecule, and tightly associated to the gelsolin-like domain of Sec23 through crystal contacts. When we included the Sec24 N-terminal 61–73 helix in the relevant pocket on sec23, this did not significantly change the positions of the fitted molecules.

We generated a model for the Sec13/31 heterotetramer from the X-ray structures of the edge and vertex elements (PDB IDs 2PM6 and 2PM9), as described in (*Fath et al., 2007*). The atomic model was initially roughly placed in the rod density, and its position automatically refined with Chimera fit command.

## Assessment of outer coat arrangement on tubes

The arrangement of the aligned outer coat vertex structure on the tubes suggests that the majority of vertices are parallel to each other, with the beta angle oriented along the tube axis, and the alpha angle around the tube. A number of observations rule out the alternative possibility that the orientation

of the vertices is random or alternate such that half the vertices are rotated by 90° to place alpha along the tube axis (this would be the case if the rods were +/−). Firstly, the final structure, obtained from vertices with similar in plane rotations within each tube, clearly shows the expected twofold features of the vertex as previously described by *Stagg et al. (2006)*, and is not a smeared average of vertices in different orthogonal orientations. Secondly, a classification of the aligned vertex subtomograms did not identify a class rotated by 90° relative to other classes. Thirdly, the measured distance between vertices along +/+ rods is as expected significantly shorter than along −/− rods, by 17 Å (*Figure 3— figure supplements 2 and 3*). Taking measurements from the fitted structures of the in vitro assembled cages one would expect a difference of approximately 40 Å, though this estimate does not take into account structural differences, between the vertices and rods of cages and those of tubes, which affect these distances. If the vertices were alternately oriented then all rods would be +/− and no difference would be expected. Equally, no difference would be expected if the vertices were randomly oriented and linked by a mixture of +/+, −/− and +/− rods. Fourthly, measuring alpha and beta based on the coordinates of the vertex and its neighbours, gave an alpha angle of 80° and a beta angle of 96°. If the vertices were alternatively oriented then no difference would be expected. Fifthly, where alpha and beta could be measured in this way at individual vertices, then beta angle was in over 90% of cases larger than alpha.

## Database depositions
A representative tomogram, the structures and the fits have been deposited in appropriate databases: EMDB accession numbers 2428, 2429, 2430, 2431, 2432; PDB accession numbers 4bzi, 4bzj, 4bzk.

# Acknowledgements
The study was technically supported by the EMBL IT services unit. RS is an investigator of the HHMI and a Senior Fellow of the University of California, Berkeley Miller Institute. The authors acknowledge Frank Thommen, Claudia Müller, Daniel Castano-Diez, and Wim Hagen for technical support, and Helen Saibil and Marco Faini for discussions.

# Additional information

### Competing interests
RS: Editor-in-Chief, *eLife*. The other authors declare that no competing interests exist.

### Funding

| Funder | Grant reference number | Author |
|---|---|---|
| Deutsche Forschungsgemeinschaft | SFB638 | Simone Prinz, John AG Briggs |
| Human Frontier Science Program | LT000978/2010-L | Giulia Zanetti |
| Bundesministerium für Bildung und Forschung | ZIK HALOmem, FKZ 03Z2HN22 | Sebastian Daum, Annette Meister, Kirsten Bacia |
| European Regional Development Fund | 1241090001 | Sebastian Daum, Annette Meister, Kirsten Bacia |
| Howard Hughes Medical Institute | | Randy Schekman |

The funders had no role in study design, data collection and interpretation, or the decision to submit the work for publication.

### Author contributions
GZ, Analysis and interpretation of data, Drafting or revising the article; SP, Performed and optimized in vitro reconstitution, Acquisition of data; SD, Performed and optimized protein expression and purification; AM, Performed and optimized in vitro reconstitution; RS, KB, Conception and design; JAGB, Conception and design, Analysis and interpretation of data, Drafting or revising the article

## Additional files

### Major datasets

**The following datasets were generated:**

| Author(s) | Year | Dataset title | Dataset ID and/or URL | Database, license, and accessibility information |
|---|---|---|---|---|
| Zanetti G, Prinz S, Daum S, Meister A, Schekman R, Bacia K, et al. | 2013 | The structure of the COPII coat assembled on membranes: Sar1/Sec23/24 inner coat | http://www.ebi.ac.uk/pdbe/entry/EMD-2428 | Publicly available at the Electron Microscopy Data Bank (http://www.ebi.ac.uk/pdbe/emdb/). |
| Zanetti G, Prinz S, Daum S, Meister A, Schekman R, Bacia K, et al. | 2013 | The structure of the COPII coat assembled on membranes: Sec13/31 outer coat vertex | http://www.ebi.ac.uk/pdbe/entry/EMD-2429 | Publicly available at the Electron Microscopy Data Bank (http://www.ebi.ac.uk/pdbe/emdb/). |
| Zanetti G, Prinz S, Daum S, Meister A, Schekman R, Bacia K, et al. | 2013 | The structure of the COPII coat assembled on membranes: Sec13/31 outer coat left-handed rod | http://www.ebi.ac.uk/pdbe/entry/EMD-2430 | Publicly available at the Electron Microscopy Data Bank (http://www.ebi.ac.uk/pdbe/emdb/). |
| Zanetti G, Prinz S, Daum S, Meister A, Schekman R, Bacia K, et al. | 2013 | The structure of the COPII coat assembled on membranes: Sec13/31 outer coat right-handed rod | http://www.ebi.ac.uk/pdbe/entry/EMD-2431 | Publicly available at the Electron Microscopy Data Bank (http://www.ebi.ac.uk/pdbe/emdb/). |
| Zanetti G, Prinz S, Daum S, Meister A, Schekman R, Bacia K, et al. | 2013 | The structure of the COPII coat assembled on membranes: representative tomogram | http://www.ebi.ac.uk/pdbe/entry/EMD-2432 | Publicly available at the Electron Microscopy Data Bank (http://www.ebi.ac.uk/pdbe/emdb/). |
| Zanetti G, Prinz S, Daum S, Meister A, Schekman R, Bacia K, et al. | 2013 | The structure of the COPII coat assembled on membranes: fit of Sar1/Sec23/24 into EMD-2428 | http://www.rcsb.org/pdb/search/structidSearch.do?structureId=4BZI | Publicly available at the Protein Data Bank (http://www.rcsb.org/pdb/). |
| Zanetti G, Prinz S, Daum S, Meister A, Schekman R, Bacia K, et al. | 2013 | The structure of the COPII coat assembled on membranes: fit of Sec13/31 into EMD-2430 | http://www.rcsb.org/pdb/search/structidSearch.do?structureId=4BZJ | Publicly available at the Protein Data Bank (http://www.rcsb.org/pdb/). |
| Zanetti G, Prinz S, Daum S, Meister A, Schekman R, Bacia K, et al. | 2013 | The structure of the COPII coat assembled on membranes: fit of Sec13/31 into EMD-2431 | http://www.rcsb.org/pdb/search/structidSearch.do?structureId=4BZK | Publicly available at the Protein Data Bank (http://www.rcsb.org/pdb/). |

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
