## [Decision Letter]

Thank you for sending your work entitled “The structure of the COPII coat assembled on membranes” for consideration at *eLife*. Your article has been favorably evaluated by a Senior editor and 3 reviewers, one of whom, Stephen Harrison, agreed to serve as a guest Reviewing editor, and one of whom, Sriram Subramaniam, has agreed to reveal his identity.

The guest Reviewing editor and the other reviewers discussed their comments before we reached this decision, and the following comments have been assembled to help you prepare a revised submission.

This analysis by electron tomography of the complete COPII coat assembled on membranes in vitro offers new insights regarding the disposition of membrane and the inner and outer COPII coat components. The difference between the coats studied in the work reported in this MS by electron cryotomography (cryoET) with subtomogram averaging and those studied previously by single-particle reconstruction is the mode of preparation. These are made by in vitro budding from GUVs; the previous coats were assembled from protein only. The new structures show that Sec13/Sec31 forms tubular lattices as well as the isometric (round) structures seen in the earlier work and that the inner protein layer (Sar1/Sec23/Sec24) packs, in these particles, into a locally ordered lattice, related to the lattice of the outer layer in the tubular regions. The main discovery is that Sec13/31 can form an extended rhomboidal lattice to coat membrane tubes – a variation on the triangle/square/pentagon polyhedra described previously. The results described in the manuscript clearly add to our knowledge of COPII coat structures, and when the manuscript is revised to meet the concerns listed below, it should meet the criteria for publication in *eLife*. The reviewing editor does wish to emphasize that he will only consider the revision if it includes real modifications that indeed respond to these concerns.

1) It is unclear how much significance should be ascribed to the Sec23/24-Sar1 inner lattice. These proteins have been driven onto membrane by the non-hydrolyzable GppNHp, resulting in their two-fold excess over Sec13/31. The authors describe tantalizing features of this regular array: the electrostatic complementarity of lattice contacts, the nice fit of modeled Sec22 SNARE, the packing of a Sec24 peptide element onto Sec23. But all this might be akin to a description of the lattice of a protein crystal. Some further discussion of this issue is needed, to make it clear that the extensive regularity might be a feature of the in vitro assembly. For example, the authors might consider to what extent cargo with appreciable cytoplasmic mass could fit into these tubular regions if they are coated with such a close-packed Sec23/24-Sar1 lattice.

A minor point in this connection was raised by one of the reviewers. The explanation that is offered for how the Sec13/31 lattice differs on tubes and spheres doesn't seem quite definitive. (The alpha and beta angles are germane, but mention of them tends to confuse as much as to resolve the issue). In the simplest sense, the extended lattice arises because the Sec13/31 rods can assemble into an array of squares (rhomboids). The +/+ and -/- arrangement of rods is not absolutely required for a rhomboidal lattice to form; a +/- arrangement would work, it would retain two-fold vertex centers. Rather, the importance of the +/+ -/- arrangement is presumably that this geometry brings the Sec13/31 lattice into close (though still imperfect) register with the inner Sec23/24-Sar1 lattice. Regarding this point, it isn't clear how continuous the +/+ -/- arrangement really is along the tubular membranes. In the Table of Figure 3—figure supplement 3, the difference between the cell edges for the left-handed (31.9 nm) and right-handed (30.2 nm) “helices” seems a bit small. It would be useful if the authors could measure this based on the Sec13/31 crystal structure. Could there in fact be occasional dislocations in the +/+ -/- lattice?

2) Tubular variants of round structures were among the earliest coat structures ever examined by electron microscopy - the polyheads of T4 and related bacteriophages, studied in detail by Kellenberger and Klug in the ’60s. Those groups examined different classes of polyheads and suggested how subunit packing might generate them. Recall also that T-even phages actually have elongated heads (because of a “waist” between the two rounded ends, each with hemi-icosahedral symmetry), and giant T4 phages have very long heads (extended tubular waists). More recently, the beautiful work of Sundquist and Finch on cone-like assemblies of the HIV-1 capsid protein illustrates an even more striking polymorphism that generates tapered tubes. So the unwarranted use at the end of the first paragraph in “Results and Discussion” of the word “unexpectedly” (an interjection that should be used sparingly, if at all, in scientific publications) is either an admission of ignorance or an unnecessary attempt to impart “novelty”. If procollagen and other elongated molecules are transported by the COPII pathway, then tubular structures with rounded ends are precisely what we can expect. The structural results presented here are interesting, not because they show tubes with hemispherical ends, but rather because the structure of the tubes is not the only one that could have been imagined from the lattice of isometric coats and because the lattice has a cogent interpretation in terms of the high resolution structures of its components. The authors should add some discussion of the clear and relevant precedents for their results.

3) The authors should pay attention to the notion of quasi-equivalence and the way in which high-resolution structures of icosahedral viruses have given this notion specific instantiations and modified earlier formulations. The authors appear not to appreciate that there is important precedent in work on virus structures, especially Klug's classic studies that followed in the initial work on polyheads. The different conformations for the edges (hinge bending) are quite consistent with similar variations in virus capsid assemblies and polymorphisms. In this respect, the description of edges as “++”, “+-”, etc. is confusing. Designations based on the contacts at the ends, rather than on the conformation of the edge dimer itself, may mislead the reader into thinking that there is some inherent asymmetry in the dimer, rather than an asymmetry in its bonding (which may or may not generate some longer-range asymmetry in its conformation).

4) The model at the end impresses two of the reviewers as unlikely or unnecessary. The packaging of an elongated cargo can drive the switch from hemispherical to tubular, just as some variation of membrane tension or other stochastic fluctuation has presumably driven the switch during the in vitro assembly of the coats studied here. Moreover, there is no reason why the outer-layer and inner-layer proteins cannot co-assemble, with the lattice of the latter imposing regularity on the packing of the former. Positing regulatory proteins that somehow communicate with cargo could send cell biologists on unnecessary wild-goose chases. In the case of elongated phage heads, an internal protein (the scaffold) appears to dictate the formation of a tubular part: it is really akin to cargo in the case of a vesicle. (The phage DNA gets stuffed in later, after the scaffold goes away.) Note that an example of a coat protein that appears not to form tubular structures (i.e., lattices curved in one direction but not the other, instead of lattices with two dimensions of curvature) is clathrin, probably because it is bent at a trimeric vertex, which can flatten (more or less) but not in one direction only. In that case, elongated cargos cause coat assembly to stall, because the shell curves back against the membrane that surrounds the cargo and cannot continue; recruitment of actin then drives further engulfment.

5) Finally, there are some technical aspects of the cryo-electron tomography that the authors should address. The information provided on the subvolume averaging is minimal, so it is not possible to assess the quality of the data. A few representative tomograms of the actual data would help. Further, the use of CTF correction is not well supported. The authors need to show power spectra without CTF correction to show that there was measurable signal beyond the first zero (for data in Figure 4). At 200 kV, and at about 2 micron defocus, the first zero crossing of the CTF is about 22.5 Å, so the claim of resolution improvement with CTF correction is hard to understand. The actual resolution is probably around 30 Å, notwithstanding the FSC plots, which are especially unreliable for defining resolution in sub-tomogram averages. Nevertheless, the mechanistic conclusions could be drawn even if the data were only at resolutions of 30-40 Å.

---

## [Author Response]

*1) It is unclear how much significance should be ascribed to the Sec23/24-Sar1 inner lattice. These proteins have been driven onto membrane by the non-hydrolyzable GppNHp, resulting in their two-fold excess over Sec13/31. The authors describe tantalizing features of this regular array: the electrostatic complementarity of lattice contacts, the nice fit of modeled Sec22 SNARE, the packing of a Sec24 peptide element onto Sec23. But all this might be akin to a description of the lattice of a protein crystal. Some further discussion of this issue is needed, to make it clear that the extensive regularity might be a feature of the in vitro assembly. For example, the authors might consider to what extent cargo with appreciable cytoplasmic mass could fit into these tubular regions if they are coated with such a close-packed Sec23/24-Sar1 lattice*.

The reviewers rightly point out that the extensive regular lattice we observe could be akin to a 2D crystal promoted by the absence of GTP hydrolysis and request further discussion of this issue. The revised manuscript now includes the text below.

In the section “The structure of the inner coat”, starting:

“We cannot rule out that the large-scale order seen in the inner coat lattice is promoted by carrying out assembly in vitro in the absence of GTP hydrolysis…”

In the section “The relationship between inner and outer coats, and implications for assembly”, starting:

“Gaps and dislocations in both inner and outer coat lattices may further contribute to curvature variability…”

*A minor point in this connection was raised by one of the reviewers. The explanation that is offered for how the Sec13/31 lattice differs on tubes and spheres doesn't seem quite definitive. (The alpha and beta angles are germane, but mention of them tends to confuse as much as to resolve the issue). In the simplest sense, the extended lattice arises because the Sec13/31 rods can assemble into an array of squares (rhomboids). The +/+ and -/- arrangement of rods is not absolutely required for a rhomboidal lattice to form; a +/- arrangement would work, it would retain two-fold vertex centers. Rather, the importance of the +/+ -/- arrangement is presumably that this geometry brings the Sec13/31 lattice into close (though still imperfect) register with the inner Sec23/24-Sar1 lattice. Regarding this point, it isn't clear how continuous the +/+ -/- arrangement really is along the tubular membranes. In the Table of*
Figure 3—figure supplement 3*, the difference between the cell edges for the left-handed (31.9 nm) and right-handed (30.2 nm) “helices” seems a bit small. It would be useful if the authors could measure this based on the Sec13/31 crystal structure. Could there in fact be occasional dislocations in the +/+ -/- lattice*?

The reviewers suggest that the difference between the +/+ and -/- edges (17Å) seems a bit small. We have measured the expected difference based on the crystal structure fitted into the cuboctahedral cage, finding it to be approximately 40Å. This measurement does not consider that changes in the vertex angles or straightening of the rods may alter the difference in length. Nevertheless, the difference between the rod lengths is indeed smaller than expected. We have included a comparison with the crystal structure.

The fact that the difference is smaller than expected may, as the reviewers suggest, result from dislocations in the lattice (we are now explicit about this in the manuscript: “*Gaps and dislocations in both inner and outer coat lattices may further contribute to curvature variability*”). The reviewers are correct that a +/- arrangement could also assemble a rhomboidal lattice, but this would require firstly that the vertex can deform to the point that alpha and beta angles are equal and secondly that it can be oriented in two ninety-degree rotated positions relative to the underlying curvature of the tube surface. In contrast, the +/+, -/- arrangement allows a rhomboidal lattice to form while maintaining a smaller alpha angle and larger beta angle at all vertices. Put more simply, assembling a rhomboidal lattice on the surface of the tube using a +/- arrangement would require a more substantial deformation of the vertex. In the original submission, the supplement contained a section on the assessment of the outer coat arrangement, where the observations indicating a predominantly “+/+” and “-/-” arrangement are described. We have edited this section (see the quote above). Further, we have clarified this issue in the main text.

We have also included an extra panel in Figure 3—figure supplement 2, with the legend:

“A rhomboidal lattice could take two forms. In the first, each vertex would have the same orientation relative to the tube (left-hand panel)…”

*2) Tubular variants of round structures were among the earliest coat structures ever examined by electron microscopy - the polyheads of T4 and related bacteriophages, studied in detail by Kellenberger and Klug in the '60s. Those groups examined different classes of polyheads and suggested how subunit packing might generate them. Recall also that T-even phages actually have elongated heads (because of a “waist” between the two rounded ends, each with hemi-icosahedral symmetry), and giant T4 phages have very long heads (extended tubular waists). More recently, the beautiful work of Sundquist and Finch on cone-like assemblies of the HIV-1 capsid protein illustrates an even more striking polymorphism that generates tapered tubes. So the unwarranted use at the end of the first paragraph in “Results and Discussion” of the word “unexpectedly” (an interjection that should be used sparingly, if at all, in scientific publications) is either an admission of ignorance or an unnecessary attempt to impart “novelty”. If procollagen and other elongated molecules are transported by the COPII pathway, then tubular structures with rounded ends are precisely what we can expect. The structural results presented here are interesting, not because they show tubes with hemispherical ends, but rather because the structure of the tubes is not the only one that could have been imagined from the lattice of isometric coats and because the lattice has a cogent interpretation in terms of the high resolution structures of its components. The authors should add some discussion of the clear and relevant precedents for their results*.

We had written: “Unexpectedly, the inner coat layer also appeared as a regular array”. To us this was indeed unexpected, but we understand that the word is ill advised and have removed it.

We thank the reviewers for suggesting a comparison of how the structural flexibility of the COPII lattice relates to previous work on elongated phage heads and HIV CA arrays. We have, as requested, added a discussion of these systems to the manuscript.

*3) The authors should pay attention to the notion of quasi-equivalence and the way in which high-resolution structures of icosahedral viruses have given this notion specific instantiations and modified earlier formulations. The authors appear not to appreciate that there is important precedent in work on virus structures, especially Klug's classic studies that followed in the initial work on polyheads. The different conformations for the edges (hinge bending) are quite consistent with similar variations in virus capsid assemblies and polymorphisms. In this respect, the description of edges as “++”, “+-”, etc. is confusing. Designations based on the contacts at the ends, rather than on the conformation of the edge dimer itself, may mislead the reader into thinking that there is some inherent asymmetry in the dimer, rather than an asymmetry in its bonding (which may or may not generate some longer-range asymmetry in its conformation)*.

We have added a comparison to virus structures (see response to the previous point.)

We appreciate the risk that the reader may be confused by the ambiguities between asymmetry in bonding and inherent asymmetry in the Sec13/31 dimers. We believe it is appropriate to adopt the same “+” and “-” notation used in the previous literature for describing the contacts at the ends of the rods, and believe that the succinct “+/-” notation for describing rods is preferable to introducing longer phrases such as “rods making a + contact at one end and a – contact at the other end”. To minimize the risk of confusion we have edited the text in two key positions:

A) “Within the in vitro assembled cages each Sec13/31 rod makes a “+” contact at one end and a “-” contact at the other end.”

B) “Together these data imply that three properties contribute to outer coat adaptability…”

*4) The model at the end impresses two of the reviewers as unlikely or unnecessary. The packaging of an elongated cargo can drive the switch from hemispherical to tubular, just as some variation of membrane tension or other stochastic fluctuation has presumably driven the switch during the in vitro assembly of the coats studied here. Moreover, there is no reason why the outer-layer and inner-layer proteins cannot co-assemble, with the lattice of the latter imposing regularity on the packing of the former. Positing regulatory proteins that somehow communicate with cargo could send cell biologists on unnecessary wild-goose chases. In the case of elongated phage heads, an internal protein (the scaffold) appears to dicate the formation of a tubular part: it is really akin to cargo in the case of a vesicle. (The phage DNA gets stuffed in later, after the scaffold goes away.) Note that an example of a coat protein that appears not to form tubular structures (i.e., lattices curved in one direction but not the other, instead of lattices with two dimensions of curvature) is clathrin, probably because it is bent at a trimeric vertex, which can flatten (more or less) but not in one direction only. In that case, elongated cargos cause coat assembly to stall, because the shell curves back against the membrane that surrounds the cargo and cannot continue; recruitment of actin then drives further engulfment*.

We agree with the reviewers that cargo packaging could drive the switch from spherical to tubular morphology, and that the inner and outer coats could co-assemble, with the inner coat imposing regularity on the outer coat, or vice versa. We regret a misunderstanding: the purpose of our model was not to posit regulatory proteins that communicate with cargo. This suggestion is in fact already prevalent in the COPII literature, and candidate proteins have already been identified. For example “… specific proteins may control the loading of big cargoes and regulate the size of COPII carriers accordingly… Recent findings have revealed that the proteins TANGO1 and cTAGE5 assemble into a dimer at the ER exit sites and that both are required for Collagen VII secretion. TANGO1 binds collagen VII and both TANGO and cTAGE5 bind COPII proteins Sec23/24… TANGO1 and its binding partners are therefore excellent candidates for factors that could delay Sar1-GTP hydrolysis in a cargo-dependent manner, by binding Sec23/Sec24 and delaying recruitment of the Sec13/Sec31 complex. In this manner, TANGOs can promote extended growth of COPII carriers.” (Malhotra and Erlmann, EMBO J., 2011) and “…there has been important progress in identifying receptors such as TANGO1 that guide the packing of PCs into COPII transport carriers… In our working hypothesis, the combined action of TANGO1 and Sedlin coordinates the stabilization of the inner COPII layer (through TANGO1) with efficient Sar1 cycling (through Sedlin) to prevent premature membrane constriction, thus allowing growth of nascent carriers and the incorporation of large PC prefibrils…” (Venditti et al., Science, 2012). Our model was intended to place our data in the context of this existing literature. Given that our data is relevant to an extensive existing discussion of carrier size modulation, we think it is appropriate to maintain some discussion of this point in the paper. Considering the concerns of the reviewers, we have edited the final paragraph of the discussion as well as the legend to Figure 7.

We hope that this new formulation of the model is a useful contribution, and does not risk sending anyone on a wild-goose chase.

*5) Finally, there are some technical aspects of the cryo-electron tomography that the authors should address. The information provided on the subvolume averaging is minimal, so it is not possible to assess the quality of the data. A few representative tomograms of the actual data would help. Further, the use of CTF correction is not well supported. The authors need to show power spectra without CTF correction to show that there was measurable signal beyond the first zero (for data in*
Figure 4*). At 200 kV, and at about 2 micron defocus, the first zero crossing of the CTF is about 22.5 Å, so the claim of resolution improvement with CTF correction is hard to understand. The actual resolution is probably around 30 Å, notwithstanding the FSC plots, which are especially unreliable for defining resolution in sub-tomogram averages. Nevertheless, the mechanistic conclusions could be drawn even if the data were only at resolutions of 30-40 Å*.

All structures, all fits, and a representative tomogram from the raw data have been deposited in the EMDB/PDB.

The range of defocuses described in the data processing section for the close-to-focus data (1.5-2.5 μm) referred to the nominal defocus at data collection, while the measured defocus of the dataset ranged from 2.0-3.2 µm. We apologize for the ambiguity, which has now been corrected.

This places the first node of the mean CTF of the data at approximately 27 Å, meaning that CTF correction is required to obtain a resolution of 20 Å.

We have now added a section to the Methods and a supplementary figure containing a validation of the importance and success of the CTF correction: we took the final reconstruction of the inner coat from CTF corrected and CTF uncorrected data, and compared it to a volume generated from the arrayed crystal structure of the Sec23/24/Sar1 complex. The FSCs between the X-ray structure and the corrected EM structure show positive correlation to a resolution of ∼20 Å, whereas the FSC between the X-ray structure and the uncorrected EM structure shows a phase inversion at approximately 27 Å as expected. This is included in Figure 4—figure supplement 1, which also includes a visualization of the map in comparison with the arrayed crystal structures filtered to the resolution of the reconstruction by multiplication with the FSC.

We note that the resolution of the map has been measured using a “gold standard” approach (the two half datasets have been processed completely independently at all stages, and a low-pass filter was applied at 45 Å during alignment). Further, the mask used for resolution measurement (which can indeed promote an over-estimate of resolution) was cautious: a sphere with a five-pixel Gaussian fall off. We have clarified this in the text.

We have added text to clarify the defocus measurement approach, where we did not make use of the image power spectra.

While we agree with the reviewer that the same conclusions could be drawn had we obtained a lower-resolution structure, we should nevertheless include our best estimate of the correct resolution. For the reasons presented above we have confidence in both the CTF correction and the resolution estimate. Within the revised main text we now use the 0.5 criterion value of 23 Å.